# Probing and manipulating the Mexican hat-shaped valence band of In$_2$Se$_3$

James Felton[1], Jordan Harknett[2], Joe Page[2], Zhuo Yang [3], Nada Alghofaili[1], James N. O'Shea [1], Laurence Eaves [1], Yoshimitsu Kohama [3], Mark T. Greenaway [2] & Amalia Patanè [1]

Ferroelectrics based on van der Waals semiconductors represent an emergent class of materials for disruptive technologies ranging from neuromorphic computing to low-power electronics. However, many theoretical predictions of their electronic properties have yet to be confirmed experimentally and exploited. Here, we use nanoscale angle-resolved photoemission electron spectroscopy and optical transmission in high magnetic fields to reveal the electronic band structure of the van der Waals ferroelectric indium selenide (α-In$_2$Se$_3$). This indirect bandgap semiconductor features a weakly dispersed valence band, which is shaped like an inverted Mexican hat. Its form changes following an irreversible structural phase transition of α-In$_2$Se$_3$ into β-In$_2$Se$_3$ via a thermal annealing in ultra-high vacuum. Density functional theory supports the experiments and reveals the critical contribution of spin orbit coupling to the form of the valence band. The measured band structure and its in situ manipulation offer opportunities for precise engineering of ferroelectrics and their functional properties beyond traditional semiconducting systems.

Semiconductors based on van der Waals crystals offer an ideal materials platform for advanced technologies[1,2]. Due to the weak bonding between van der Waals layers, it is possible to isolate and then re-assemble atomically-thin layers without the constraints that apply to traditional semiconductors. In particular, reducing the dimensionality of a van der Waals crystal induces qualitative changes in the electronic band structure and, correspondingly, in electronic properties and functionalities[3–6]. One such modification is the emergence of a valence band (VB) shaped like an inverted Mexican hat (IMH). This is accompanied by van Hove singularities (vHs) in the density of states (DOS) and hole effective masses that are heavier than in traditional covalent semiconductors, such as Si[7–9]. The interest in this unusually shaped band and heavy carrier masses is motivated by the possibility to create new forms of charge, magnetic, and superconducting order, driven by weakly screened electron correlations[3,9–12].

The family of van der Waals crystals presently comprises over 1000 materials[13]. Yet, only a few of these are semiconductors with an IMH-shaped valence band. These include the metal chalcogenide (MC)

compounds (M = group-III metals Ga and In; C = group-VI chalcogens S, Se and Te), such as single or a few-layer InSe and GaSe[4,14–16]. Most of these materials recover a conventional parabolic valence band as the layer thickness is increased above a few nanometres. On the other hand, theoretical predictions of the band structure of the MC In$_2$Se$_3$ show distinct band structures[17–19] and the presence of an IMH-shaped VB in single and/or bulk layers[17–19]. Since In$_2$Se$_3$ possesses intrinsic ferroelectricity in both single and multiple-layers[20–22], and can exist in different polymorphs (α and β) and stacking arrangements of the van der Waals layers (1 T, 2H, and 3 R)[23], the fine-tuning of its band structure has great potential for the development of systems with ferroic order (e.g. multiferroics with ferromagnetism, ferroelectricity, and ferroelasticity)[17,18]. This could be realised by combining the IMH-shaped VB with a precise tuning of the Fermi level to the vHs in the DOS[24]. However, theoretical predictions for these new band structures and phenomena are yet to be realised experimentally.

Measurements of electronic band structures can be conducted by angle-resolved photoemission spectroscopy (ARPES)[24–27]. For ARPES

[1]School of Physics and Astronomy, University of Nottingham, Nottingham, UK. [2]Department of Physics, Loughborough University, Loughborough, UK. [3]Institute for Solid State Physics, The University of Tokyo, Kashiwa, Chiba, Japan. ✉e-mail: m.t.greenaway@lboro.ac.uk; amalia.patane@nottingham.ac.uk

studies, a sample must be crystalline with a clean surface. Despite attempts to measure the valence band of $In_2Se_3$ by ARPES[24–26], the electronic properties of this unique family of semiconductors are still unknown and often masked by surface effects. For example, an unusual phenomenon was observed in bulk $\alpha$-$In_2Se_3$, which is the occurrence of a metallic two-dimensional electron gas at the surface of vacuum-cleaved single crystals[27]. Here, we reveal the electronic band structure of the $\alpha$-phase of bulk $In_2Se_3$ using nanoscale ARPES and optical transmission in high magnetic fields (up to 60 T). We demonstrate unique band structure properties of bulk $\alpha$-$In_2Se_3$. This is an indirect band gap semiconductor with an IMH-shaped valence band that features a hexagon of six VB maxima (VBM) alternated to six saddle points. The valence band is modified and probed following controlled changes of the polymorph and stacking of the van der Waals layers by a thermal annealing in ultra-high vacuum (UHV). We induce an irreversible structural phase transition of $\alpha$-$In_2Se_3$ into $\beta$-$In_2Se_3$, and model the dependence of the IMH valence band on the specific polymorph ($\alpha$ or $\beta$) and layer stacking (1 T, 2H, and 3 R) by density functional theory (DFT) with the inclusion of spin orbit coupling (SOC). The data serve as a test bed of theoretical models in the literature and inform further developments of this emergent class of materials, which require precise engineering of electronic properties for a myriad of potential applications ranging from ferroelectricity-based nanoelectronics to non-volatile memories[28] and phase-change memory devices[29–31].

## Results and discussion

### Inverted Mexican hat valence band in 2H-$\alpha$-$In_2Se_3$

We first focus on 2H-$\alpha$-$In_2Se_3$, which crystallises in the $P63mc$ space group with lattice parameters $a = b = 4.05$ Å, $c = 18.75$ Å, and a unit cell comprising two van der Waals layers[23]. For our study, thin films were exfoliated from single crystals of 2H-$\alpha$-$In_2Se_3$ and transferred onto a gold substrate by visco-elastic stamping (see details in the "Methods" section). Here, we consider films with thickness $t = 0.3$ μm and 0.4 μm, as measured by atomic force microscopy, AFM (Supplementary Figs. 1 in Supplementary Note 1). Previous studies of thin films of $\alpha$-$In_2Se_3$ by piezoresponse force microscopy (PFM) confirmed their room-temperature ferroelectricity[31].

For this work, the samples were annealed in UHV at a temperature of up to $T_a = 180$ °C for 1 h to produce a clean surface for the photo-emission electron spectroscopy studies. Following the annealing, the samples were transferred via an UHV transfer line to a chamber for electron spectroscopy for chemical analysis (ESCA). All studies were conducted using an ultraviolet source at constant photon energy of 21.219 eV. The specific polytype phase of the crystal and its stability in air were assessed before and after annealing by Raman spectroscopy at several positions on the sample (Supplementary Figs. 2 in Supplementary Note 2 and "Methods").

Figure 1a shows the unit cell, the Brillouin zone (BZ) and its hexagonal in-plane projection for bulk 2H-$\alpha$-$In_2Se_3$. Figure 1b shows the room temperature ARPES constant energy surfaces for 2H-$\alpha$-$In_2Se_3$. The projected in-plane hexagonal BZ is overlayed on one of the plots, showing the $\bar{K}$ and $\bar{M}$ points at $k$-vectors of 1.0 Å$^{-1}$ and 0.9 Å$^{-1}$, respectively (see the constant energy surface at 2.22 eV below the Fermi level in Fig. 1b). Here, the bar notation denotes the in-plane projection over the volume sampled in the BZ. The ARPES studies with high $k$-resolution (Fig. 1c) show that the upper part of the VB takes the form of an IMH with a hexagon of six VBM located at $(0.25 \pm 0.01)$ Å$^{-1}$ along the $\bar{\Gamma} \to \bar{M}$ directions. To illustrate more clearly the form of the VB, we consider the energy contour plots (Figs. 1d and 1e, top panels), as derived from an edge-fitting algorithm that determines the onset of the measured photoelectron emission. The plots reveal hexapetalus flower-shaped bands with characteristic spoke-like features extending towards the $\bar{M}$ points. The energy difference between the VB edge at $\bar{\Gamma}$ and the VBM is $\Delta_{IMH} = (85 \pm 10)$ meV. Along the rim of the IMH-shaped valence band, the six VBM are separated by saddle points (SP) with an energy difference between each VBM and saddle point of $\Delta_{VBM-SP} = (29 \pm 10)$ meV. The experiments are reproduced by DFT, as shown in Fig. 1d–e (bottom panels).

In the ARPES experiment, the translational symmetry is broken by the interface of the sample with the vacuum. Thus, the out-of-plane wave vector ($k_z$) of the electron is not conserved and the ARPES data give the binding energy of the electron and the corresponding component of its wave vector parallel to the sample surface. The weakly dispersed IMH-shaped valence band seen in the experiment (Figs. 1e

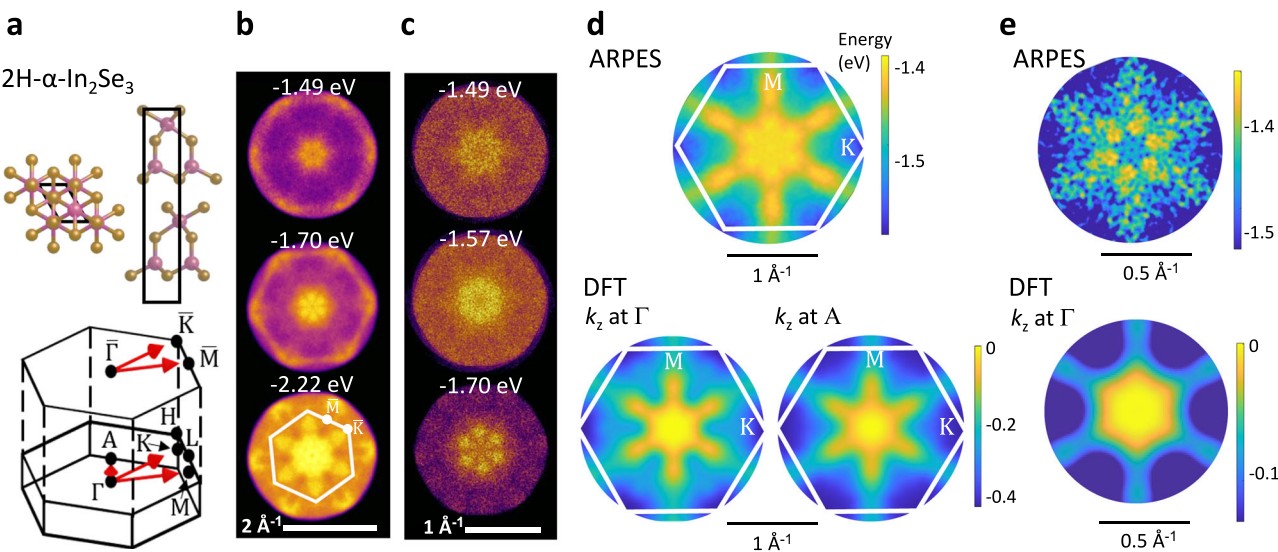

**Fig. 1 | Hexapetalus flower-shaped bands in bulk 2H-$\alpha$-$In_2Se_3$. a** Side and in-plane view of a single van der Waal layer ($\alpha$-$In_2Se_3$) and the full and projected Brillouin zone (BZ) for bulk 2H-$\alpha$-$In_2Se_3$. **b** Constant energy ARPES slices taken near the valence band maximum, VBM (photon energy 21.219 eV). **c** Constant energy ARPES slices taken near the VBM with high $k$-resolution. **d** Colour plot of the electron energy (in eV) versus in-plane $k$-vector, as determined by ARPES (top) and DFT (bottom). The DFT plots correspond to different out-of-plane wave vectors $k_z$ (at the centre, $\Gamma$, and at the edge, A, of the BZ). The projected in-plane hexagonal BZ is overlaid on the plots. **e** Colour plot of the electron energy (in eV) versus in-plane $k$-vector, as determined by ARPES (top) and DFT at $\Gamma$ (bottom). Experimental data in (e) were acquired with greater $k$-resolution and over a narrower energy range than in part (d). Energies (in eV) are referenced to the Fermi level for ARPES and to the VBM for DFT.

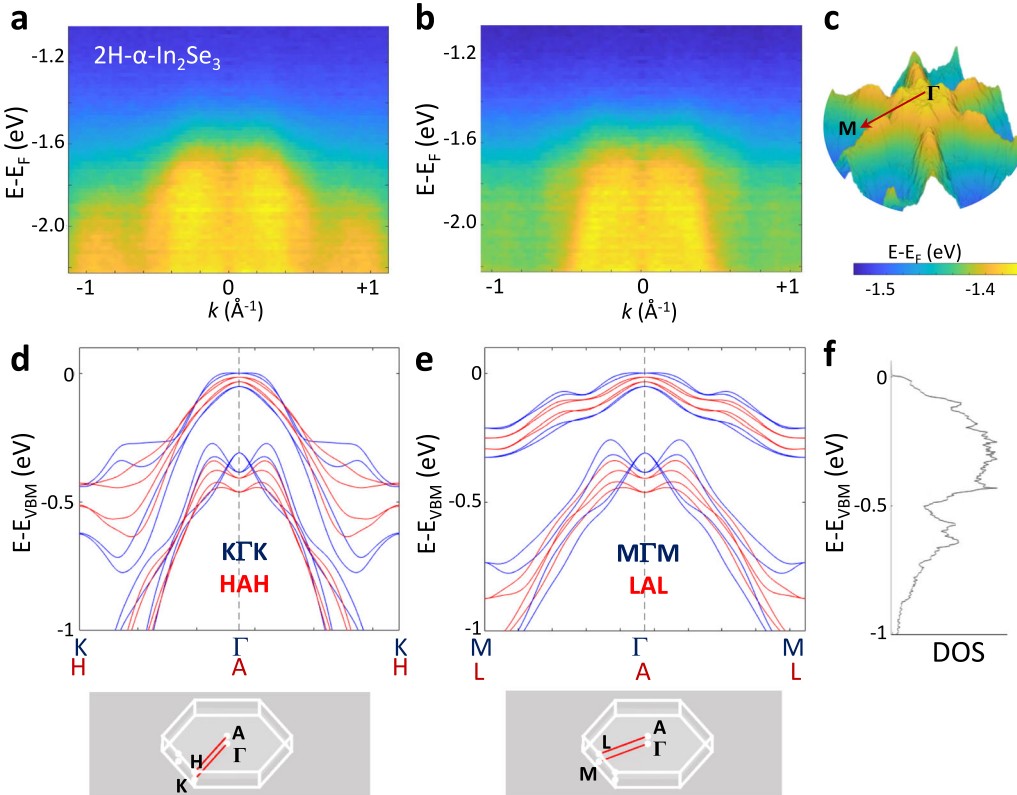

**Fig. 2 | Inverted Mexican hat valence band in bulk 2H-α-In₂Se₃. a, b** ARPES data for bulk 2H-α-In₂Se₃ taken along the high symmetry (**a**) Γ̄ → K̄ and (**b**) Γ̄ → M̄ directions. **c** Energy dispersion as measured by ARPES for the uppermost VB. Energies (in eV) are referenced to the Fermi level. **d, e** Energy bands as determined by DFT through the centre and edge of the BZ. Lower panels: BZ and high symmetry directions. **f** Calculated density of states (DOS) versus energy.

## Table 1 | Calculated and experimental carrier effective masses at selected high symmetry points in the conduction band (CB) and valence band (VB), and their geometric average, for bulk 2H-α-In₂Se₃

| Theory | $m_{\|\Gamma\text{-K}}/m_e$ | $m_{\|\Gamma\text{-M}}/m_e$ | $(m_{\|\Gamma\text{-K}} m_{\|\Gamma\text{-M}})^{1/2} /m_e$ |
|---|---|---|---|
| VB$_\Gamma$ | (−0.66) −2.60 | (−0.66) −2.59 | (2.00) 2.59 |
| VBM | (0.43) 1.65 | (0.96) 1.31 | (0.64) 1.47 |
| CBM | (0.10) 0.11 | (0.10) 0.13 | (0.10) 0.12 |
| **Experiment** | $m_{\|\Gamma\text{-K}}/m_e$ | $m_{\|\Gamma\text{-M}}/m_e$ | $(m_{\|\Gamma\text{-K}} m_{\|\Gamma\text{-M}})^{1/2} /m_e$ |
| VB$_\Gamma$ | −0.87 ± 0.17 | −0.70 ± 0.11 | 0.78 ± 0.20 |
| VBM | 1.46 ± 0.20 | 0.99 ± 0.14 | 1.20 ± 0.24 |
| CBM | 0.12 ± 0.02 | 0.12 ± 0.02 | 0.12 ± 0.03 |

Here, CBM and VBM correspond to the CB minimum and VB maximum, respectively. The values in brackets correspond to the masses calculated without the inclusion of spin-orbit coupling (SOC).

(top panel) and Fig. 2a–c) is confirmed by the calculated energy bands (Fig. 1e (bottom panel) and Fig. 2d–e). In comparing the ARPES with the DFT data, we consider the calculated dispersions for different $k_z$, at the centre (Γ) and at the edge of the BZ, i.e. along KΓK and HAH (Fig. 2d) and along MΓM and LAL (Fig. 2e). Spin-orbit coupling (SOC) is taken into account (Supplementary Figs. 3 in Supplementary Note 3). The SOC tends to flatten the bands at the Γ point and plays a crucial role in determining the form of the valence band. The DFT calculations reveal that the VBM occurs at $k_z = 0$ and that the IMH-shaped VB is preserved for different $k_z$ (Figs. 1d, Supplementary Fig. 4 in Supplementary Note 3, and Supplementary Movie 1). Thus, disorder and electron-phonon interactions at room temperature do not prevent the observation of this unusually shaped and weakly dispersed VB.

We use a parabolic fit to the measured energy dispersions (Supplementary Fig. 5 in Supplementary Note 4) to estimate the hole effective mass. At the Γ point, the hole effective mass is negative: $m_{\|\Gamma\text{-K}} = (−0.87 ± 0.17)m_e$ and $m_{\|\Gamma\text{-M}} = (−0.70 ± 0.11)m_e$ along the Γ̄ → K̄ and Γ̄ → M̄ directions, respectively. Conversely, for the VBM we obtain a positive hole effective mass with $m_{\|\Gamma\text{-M}} = (0.99 ± 0.14)m_e$ and $m_{\|\Gamma\text{-K}} = (1.46 ± 0.20) m_e$ along Γ̄ → M̄ and along Γ̄ → K̄, respectively. The measured and DFT values are listed in Table 1. Here, we compare the calculated masses with and without the inclusion of SOC. The SOC plays an important role in shaping the VB, energy dispersions and values of the hole effective masses. The effective mass for holes is significantly larger ($\approx m_e$, where $m_e$ is the electron mass in vacuum) than for electrons ($\approx 0.1m_e$,) and changes its sign going from the VBM to the saddle points on the brim of the inverted Mexican hat VB. This accounts for a steep increase in the DOS, as shown in Fig. 2f.

### Interband transitions in 2H-α-In₂Se₃

Bulk 2H-α-In₂Se₃ is an indirect bandgap semiconductor with a conduction band minimum (CBM) centred at Γ (Fig. 3a). The low energy part of the CB has a simple parabolic form with a light electron effective mass ($m_{\|\Gamma\text{-K}} = 0.11 m_e$ and $m_{\|\Gamma\text{-M}} = 0.13 m_e$, Table 1). Thus, the lowest energy interband transitions involve electrons at Γ and holes at the VBM ($E_{i1}$) and saddle points on the rim of the IMH valence band ($E_{i2}$) (arrows in Fig. 3a). To measure these transitions, we have conducted magneto-optical transmission experiments. The geometry of our pulsed magnetic field experiments (with **B** up to 60 T) is restricted to the Faraday configuration (**B** is perpendicular to the optical electric field **E** and to the layer plane of the crystal). Figure 3b shows the optical transmission spectra at **B** = 0 T and at temperatures ranging from T = 4.2 K to 293 K. Here, the indirect absorption edge shifts from $h\nu =$

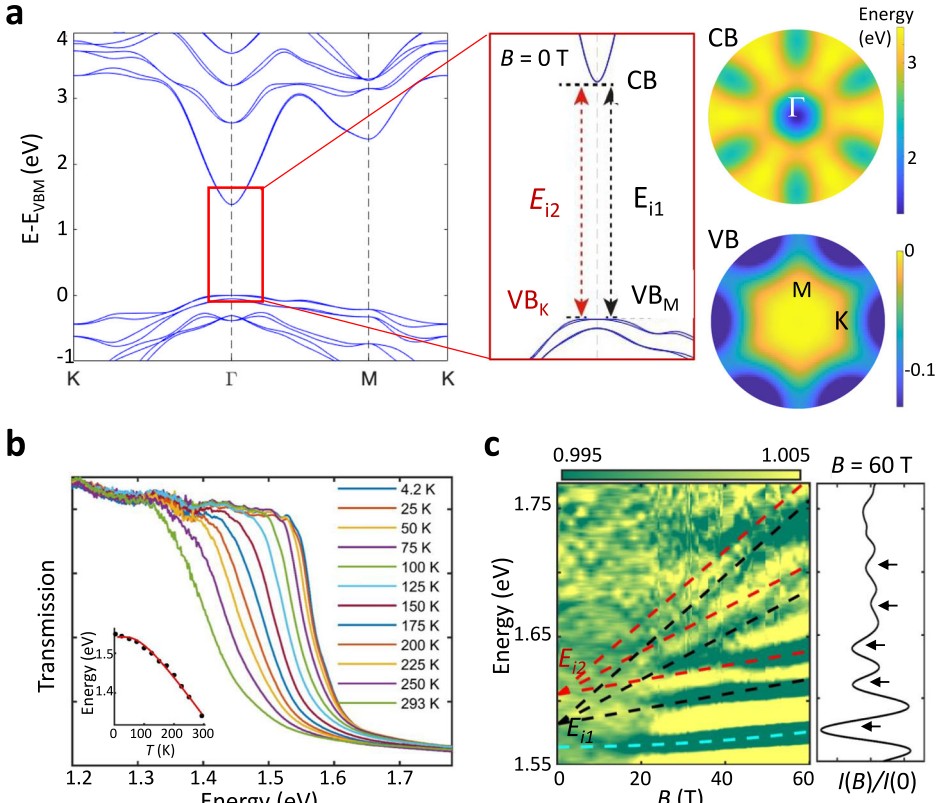

**Fig. 3 | Magneto-optical transmission of bulk 2H-α-In₂Se₃. a** Band structure, interband transitions and colour plots of the electron energy versus in-plane $k$-vector with valence band (VB) and conduction band (CB). **b** Optical transmission versus photon energy at different temperatures $T$ (magnetic field $B = 0$ T). Inset: Band edge absorption energy as a function of $T$, as derived from the crossing point of the tangent line along the band edge absorption and the background signal. The line is a fit to the data by an empirical law, as described in the Supplementary Note 5.

**c** Colour plot of the magneto-transmission (normalised to the transmission at $B = 0$ T) versus $B$ and photon energy ($T = 4.2$ K). Dashed lines show the calculated exciton (cyan) and electron-hole interband Landau level transitions (red and black lines). Lines correspond to transitions from the VBM to the CBM (black) and from the saddle points in the VB to the CBM (red lines). Right: Line profile of the magneto-transmission spectrum at $B = 60$ T. Arrows mark resonant features.

1.35 eV at $T = 293$ K to $h\nu = 1.54$ eV at $T = 4.2$ K. These energies compare favourably to the experimental values reported in the literature, which range from ~ 1.2 eV to 1.58 eV at room temperature[26,32–35]. However, previous works report that bulk α-In₂Se₃ is a direct band gap semiconductor[27,32]. Here, we use magneto-transmission to identify the band edge absorption and its nature.

Figure 3c shows a colour plot of the low temperatures ($T = 4.2$ K) magneto-optical transmission spectrum normalised to the transmission at $B = 0$ T (*e.g.* plots of $I(B)/I(0)$ versus $B$ and photon energy). At $B = 60$ T five strong absorption features can be clearly seen (black arrows in Fig. 3c). The lowest energy transition ($E_{ex}$) is observed only at high $B$ and assigned to an excitonic absorption. Its $B$-dependence is described by $E_{ex} = E_{i1} - R_y^* + \sigma B^2$, where $R_y^*$ is the exciton binding energy at $B = 0$ T and $\sigma$ is the diamagnetic shift coefficient[36]. The best fit to the data is obtained with $R_y^* = 18$ meV and $\sigma = 3.3$ μeV T⁻² (cyan line in Fig. 3c). The other prominent resonant features are attributed to interband Landau level (LL) transitions with a $B$-dependent energy shift $\Delta E = (n + \frac{1}{2})\hbar\omega_c$, where $n$ is the LL number, $\hbar$ is reduced Planck's constant, $\omega_c = eB/\mu$ is the cyclotron frequency, $\mu = (m_h^* m_e^*)/(m_h^* + m_e^*)$ is the reduced cyclotron mass due to CB electrons ($m_e^*$) and VB holes ($m_h^*$), and $e$ is the electron charge[37]. Two series of LLs can be clearly identified: the LL series with the lowest energies (black lines in Fig. 3c) is assigned to the indirect transition from the VBM to the CBM ($E_{i1}$) with $\mu = (0.11 \pm 0.01)\,m_e$. From the measured (ARPES) values of the hole mass for the VBM, we estimate $m_e^* = (0.12 \pm 0.02)\,m_e$. Likewise, for the second series of LLs due to the indirect transition from the saddle points in the VB to the CBM, $E_{i2}$ (red lines in Fig. 3c), we derive

$\mu = (0.11 \pm 0.01)\,m_e$. Using the DFT value for the hole effective mass at the saddle points, we obtain $m_e^* = (0.13 \pm 0.02)\,m_e$, in good agreement with the DFT value (Table 1); also, at $B = 0$ T, the energy difference between the two indirect transitions, $E_{i1} - E_{i2} = 23$ meV, is in agreement with the difference between the VBM and the saddle points measured by ARPES: $\Delta_{VBM-SP} = (29 \pm 10)$ meV.

In summary, the magneto-optical transmission experiments support the ARPES data and show that 2H-α-In₂Se₃ is an indirect band gap semiconductor featuring two closely spaced interband optical transitions. The LL quantisation of these transitions indicates that the CB electrons are much lighter than the VB holes, leading to weakly bound excitons and to an electron cyclotron energy that is approximately 10 times larger than for VB holes.

## Band structure engineering by a phase transition from α- to β-phase

The transition from α- to the β-In₂Se₃ is known to feature a change in the bond lengths and layer stacking configurations[38,39]; the corresponding increase in electrical conductivity has attracted interest for optical data storage and electrical non-volatile memories[39,40]. However, the measurement and manipulation of the electronic band structure by a phase transition has never been reported. This is achieved here by a thermal annealing in UHV of α-In₂Se₃, avoiding contamination and degradation of its surface in air[41].

The ARPES studies were conducted at room temperature following repeated thermal annealing in UHV of the α-In₂Se₃ crystal at temperatures ranging from $T_a = 100\,°C$ to 240 °C. After each annealing,

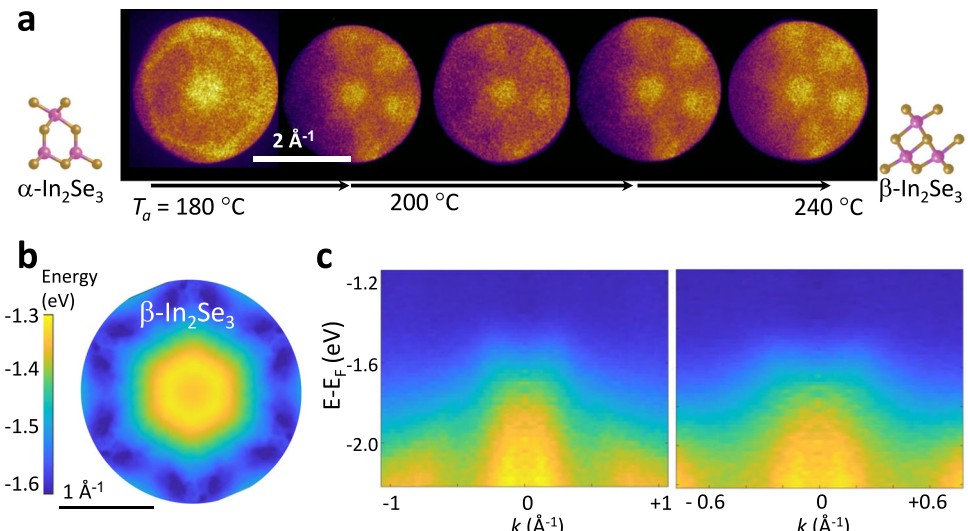

**Fig. 4 | Phase change from α-to β-In₂Se₃. a** ARPES constant energy slices showing the transition from α to β following an annealing at temperatures $T_a$ above 180 °C. The insets show the side view of the single layer for the α and β phases. All measurements were conducted at room temperature. **b** Colour plot of the electron energy (in eV) versus $k$-vectors, as determined by ARPES for bulk β-In₂Se₃. Energies (in eV) are referenced to the Fermi level. **c** ARPES data for bulk β-In₂Se₃ taken along high symmetry directions.

changes to the sample were monitored at room temperature by ARPES (Fig. 4a), energy-filtered photoemission electron microscopy (PEEM) and ultraviolet photoelectron spectroscopy (UPS) (Supplementary Fig. 6 in Supplementary Note 6). Following the electron spectroscopy studies, the phase change from α- to β-In₂Se₃ and its stability in air were also probed by Raman spectroscopy at several positions of the sample (Supplementary Fig. 2).

As shown in Fig. 4a, the ARPES constant energy surfaces change following a thermal annealing of 2H-α-In₂Se₃ at $T_a > 180$ °C. The PEEM images reveal an inhomogeneous change across the surface, which comprises of dark and bright regions (Supplementary Fig. 6 in Supplementary Note 6); also, plots of the measured work function versus $T_a$ reveal a change of up to ~0.3 eV across the flake, with slightly larger increases in the dark regions of the PEEM images and the emergence of wrinkle-like features on the flake surface. Despite β-In₂Se₃ is more defective than α-In₂Se₃, we can resolve its IMH-shaped valence band and its sixfold symmetry (Fig. 4b–c). The measured width of the IMH valence band is of comparable magnitude in both crystals. Also, the measured values of the work function and VBM, and their comparison with the band gap energies indicate that both α-In₂Se₃ and β-In₂Se₃ are $n$-type semiconductors, in line with previous reports of $n$-type conductivity in nominally undoped α- and β-In₂Se₃[42,43].

The β-phase of In₂Se₃ can exist in three different stacking configurations, 1 T, 2H, and 3 R, with *P-3m1*, *P63mc*, and *R-3m* space groups, respectively. The BZ for the 1 T and 2H phases is a hexagonal primitive cell, differing from the rhombohedral cell of the 3 R phase, as shown in Figure 5a[23]. Figure 5b shows the calculated colour plots of the electron energy versus in-plane wave vector for the uppermost electronic valence bands for each layer stacking of β-In₂Se₃ (see also Supplementary Fig. 7 in Supplementary Note 7). These all take on a similar form, with a sixfold symmetry and six VBM located away from the Γ-point. The VB is shaped like an IMH in all phases, as also observed in the ARPES data for samples annealed at $T_a = 240$ °C (Fig. 4b). To correctly identify the phase in the experiment, the measured $k$-position of the IMH brim was compared to the theoretical values for the 1 T, 2H, and 3 R phases. We find that the calculated bands for the 2H and 3 R phases provide the best description of the ARPES data.

From parabolic fits to the measured dispersions (Supplementary Figs. 8 in Supplementary Note 8), we estimate that at the $\bar{\Gamma}$ point the

hole effective mass is $m_{\parallel\bar{\Gamma}\text{-}\bar{K}} = (-4.3 \pm 0.6)m_e$ and $m_{\parallel\bar{\Gamma}\text{-}\bar{M}} = (-2.8 \pm 0.2)m_e$ along the $\bar{\Gamma} \rightarrow \bar{K}$ and $\bar{\Gamma} \rightarrow \bar{M}$ directions, respectively. Similarly, at the VBM, $m_{\parallel\bar{\Gamma}\text{-}\bar{K}} = (4.0 \pm 0.2)m_e$ and $m_{\parallel\bar{\Gamma}\text{-}\bar{M}} = (2.6 \pm 0.2)m_e$ along the $\bar{\Gamma} \rightarrow \bar{K}$ and $\bar{\Gamma} \rightarrow \bar{M}$ directions, respectively. These values are significantly larger than those predicted by theory for β-In₂Se₃ (Table 2). We assign this discrepancy to a broadening of the measured energy dispersions, which arises from the increased disorder following the transition of the crystal from α- to β-In₂Se₃.

The β-phase of In₂Se₃ has been proposed as a pseudo-centrosymmetric $β_{pc}$-phase due to the random off-centre positions of the Se-atoms[44]. The random displacement of the Se-atoms creates an effective energy barrier that can prevent a reversible transformation of the crystal from the β- to the α-phase as the annealing temperature is lowered to room temperature. In general, the crystallisation behaviour of In₂Se₃ under different conditions (*e.g.* temperature, pressure, etc.) can vary. For example, it has been reported that β-In₂Se₃ is stable in thin films or nanosheets at room temperature[45] and that the β-phase can persist in bulk crystals at room temperature in non-oxidative environments[46].

For all stacking configurations of the β-phase and in contrast to the α-phase, the CBM is shifted away from Γ (Figs. 5c and Supplementary Fig. 7). Also, while the energy dispersions around Γ are isotropic, they are anisotropic at the VBM and CBM (Figs. 5 and Table 2). Thus, the phase change from α to β creates a semiconductor with a distinct band structure and electronic properties. Finally, we note that DFT predicts a distinct shape of the valence band in single layers compared to bulk crystals: DFT predicts an IMH valence band only without the inclusion of SOC (Supplementary Figs. 9 in Supplementary Note 9). Taking into account SOC, the valence band maximum shifts to Γ in both single layers of α- and β-In₂Se₃. Thus, an indirect-direct bandgap transition is expected in α-In₂Se₃ going from bulk to single layers, which requires experimental evidence. The strong influence of SOC on the form of the valence band is in stark contrast with that of other metal chalcogenides, such as InSe and GaSe, in which SOC coupling tends to be weaker, and an IMH valence band is predicted and observed in single layers[4,14–16,47].

In summary, we have probed directly the electronic band structure of bulk α-In₂Se₃, an indirect band gap ferroelectric semiconductor with an inverted Mexican hat-shaped valence band, very weakly dispersed energy bands and heavy hole effective masses. We have shown

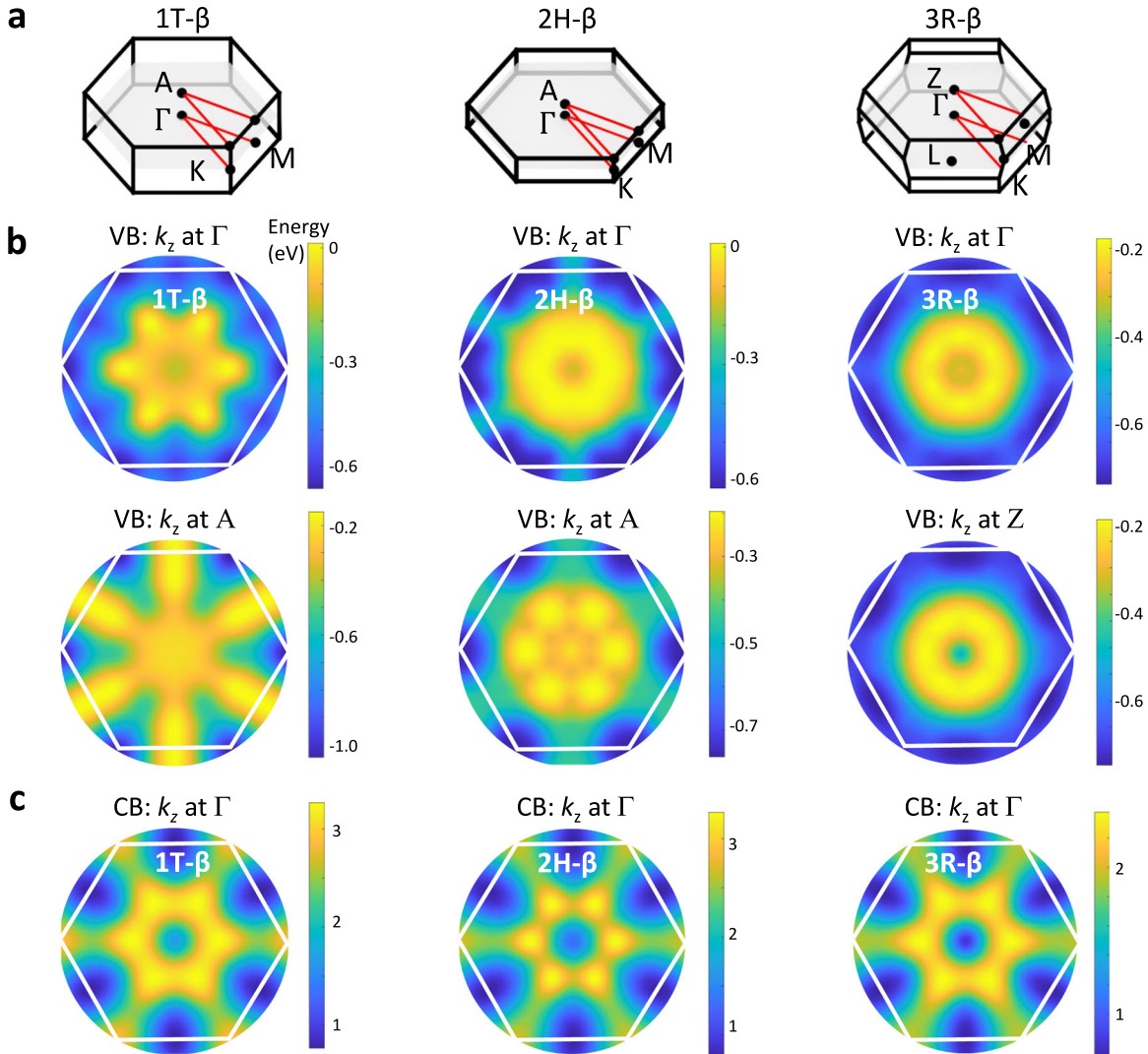

**Fig. 5 | Band structure and polytypes of bulk β-In₂Se₃. a** The Brillouin zones (BZ) of 1 T, 2H and 3 R β-In₂Se₃. **b** Calculated colour plot of energy (in eV) versus in-plane $k$-vector for the valence band of bulk 1 T, 2H and 3 R β-In₂Se₃ for $k_z$ at Γ and at the edge of the BZ. The zero energy corresponds to the valence band maximum.

**c** Calculated colour plot of energy (in eV) versus in-plane $k$-vector of the conduction band of bulk 1 T, 2H and 3 R β-In₂Se₃. The zero energy corresponds to the conduction band minimum. The projected in-plane hexagonal BZ is overlayed on the plots in parts (**b**) and (**c**).

that the phase-change from α- to β-In₂Se₃ via a thermal annealing in UHV can be irreversible and can provide a controlled means of modifying the electronic band structure. The interest in inverted Mexican hat valence bands is motivated by the possibility of creating new forms of charge, magnetic, and superconducting order, driven by electron correlations, as observed in twisted bilayer graphene[48,49], but with the additional benefit of a tuneable ferroelectric polarisation. Resonances in the DOS, as predicted for In₂Se₃, are central to the investigation of topological changes of the Fermi surface and carrier correlations; also, they are of interest for precision engineering of electronic properties and the rational design of future transistors beyond traditional technologies[50]. In particular, SOC are significant in In₂Se₃, implying a strong influence of an applied electric field on the band structure with prospects to generate spin currents and polarisation via SOC.

Future advances require precise manufacturing technologies. The controlled doping of the layers remains an outstanding challenge and requires techniques to achieve sufficient carrier densities for tuning the Fermi level to the valence band. This could be achieved by chemical and electrostatic doping, and/or photodoping. Growth under metal-rich conditions to produce Se-vacancies and doping with group-V atoms to fill the vacancies[28] or Sb-doping[51] can provide an effective

doping strategy. Also, growth and in situ characterisation tools are pivotal to developing atomically thin semiconductors with precisely engineered physical properties. The latter can be affected by exposure of surfaces to chemical species in the air, such as oxygen. Thus, the integration of growth and electron spectroscopy in UHV would be required to overcome these limitations[47]. Further opportunities may also emerge from other synthetic techniques, as recently demonstrated for subnanometre-wide In₂Se₃[52], guiding further developments of ferroelectrics.

## Methods

### Materials

Samples of commercially available bulk 2H-α-In₂Se₃ were obtained from 6Carbon Technology grown using chemical vapour transport. The mechanical exfoliation of the crystal was conducted using a Loadpoint LP:5660 blue tape with a PVC base and acrylic adhesive. The exfoliated flakes of In₂Se₃ were then stamped directly onto Au substrates (40 nm thick) prepared via thermal evaporation. Within 15 minutes of the initial stamping, the samples were placed under vacuum ($< 10^{-10}$ mbar). Annealing was conducted under vacuum via a resistive heater and monitored with an in-situ thermocouple.

**Table 2 | Calculated and experimental carrier effective masses at selected high symmetry points in the valence band (VB), and their geometric average, for bulk 1 T, 2H and 3 R β-In$_2$Se$_3$**

| 1T-β Theory | $m_{\parallel\Gamma\text{-}K}/m_e$ | $m_{\parallel\Gamma\text{-}M}/m_e$ | $(m_{\parallel\Gamma\text{-}K}\, m_{\parallel\Gamma\text{-}M})^{1/2}/m_e$ |
|---|---|---|---|
| VB$_\Gamma$ | −1.50 | −1.50 | 1.50 |
| VBM | 1.20 | 1.20 | 1.20 |
| **2H-β Theory** | $m_{\parallel\Gamma\text{-}K}/m_e$ | $m_{\parallel\Gamma\text{-}M}/m_e$ | $(m_{\parallel\Gamma\text{-}K}\, m_{\parallel\Gamma\text{-}M})^{1/2}/m_e$ |
| VB$_\Gamma$ | −0.80 | −0.80 | 0.80 |
| VBM | 1.60 | 1.00 | 1.26 |
| **3R-β Theory** | $m_{\parallel\,\Gamma\text{-}K}/m_e$ | $m_{\parallel\Gamma\text{-}M}/m_e$ | $(m_{\parallel\,\Gamma\text{-}K}\, m_{\parallel\Gamma\text{-}M})^{1/2}/m_e$ |
| VB$_\Gamma$ | −0.85 | −0.85 | −0.85 |
| VBM | 1.10 | 0.35 | 0.62 |
| **3R-β Experiment** | $m_{\parallel\Gamma\text{-}K}/m_e$ | $m_{\parallel\Gamma\text{-}M}/m_e$ | $(m_{\parallel\Gamma\text{-}K}\, m_{\parallel\Gamma\text{-}M})^{1/2}/m_e$ |
| VB$_\Gamma$ | −4.3 ± 0.6 | −2.8 ± 0.2 | 3.5 ± 0.5 |
| VBM | 4.0 ± 0.2 | 2.6 ± 0.2 | 3.2 ± 0.3 |

Here, VBM correspond to the VB maximum.

## Electron spectroscopy

ARPES and ultraviolet photoelectron spectroscopy (UPS) measurements were conducted using a Scienta Omicron ESCA equipped with a Focus HIS 14 HD UV photon source with energy $h\nu = 21.219$ eV. This is well suited for UPS experiments, which require a small excitation area and high photon flux. The ESCA was operated in energy filtered mode with a pass energy of 50 eV and slit width of 0.2 mm, leading to an energy resolution of 40 meV. For ARPES, photoelectrons were collected from a region of approximately $10 \times 20\ \mu m^2$. The experiments were conducted over several areas of each sample to probe its uniformity. Both ARPES and UPS were conducted with the photoemission electron microscopy (PEEM) optics in telescopic mode with an extractor voltage of 12000 V. The Fermi energy, energy resolution and $k$ calibration were confirmed through measurements on a gold reference sample. The measured constant energy slices were rotationally symmetrised around the $\bar{\Gamma}$ point with a six fold symmetry. This improved the signal to noise ratio and reduced the effect of non-uniform detector illumination, thus improving the contrast.

## Magneto-transmission spectroscopy

For magneto-optical transmission experiments, samples of In$_2$Se$_3$ were placed in a liquid helium cryostat. A non-destructive magnet was used to generate magnetic fields up to 60 T with a pulse duration of 36 ms in the Faraday configuration (**B**⊥**E**). White light was directed to the sample using a 400 μm diameter optical fibre with a broadband halogen lamp acting as the light source. The transmitted light was collected by a 800 μm diameter fibre and guided to a spectrometer equipped with a charge-coupled device (CCD) camera. The typical exposure time was 0.5 ms, which ensured the transmission spectra were obtained at an essentially fixed magnetic field.

## Raman spectroscopy

Raman spectroscopy was conducted using a Horiba LabRAM HR utilising a He-Ne laser ($\lambda = 632.8$ nm) and a 1200 g mm$^{-1}$ diffraction grating. Light was directed to and from the sample through a confocal microscope equipped with a 100× objective (laser spot size of $\approx 1\ \mu m$) allowing for the flakes in question to be located with the aid of a motorised sample stage. The dispersed Raman signal was detected using a CCD camera. The Raman spectra were acquired over several spots of each sample to probe its uniformity. For our relatively thick samples, the Raman spectra are dominated by bulk properties.

## Atomic force microscopy

Atomic force microscopy was conducted in air using an Asylum research Cypher-S AFM system operating in non-contact mode.

## Density functional theory

The density functional calculations, which include spin-orbit coupling, were undertaken using the Perdew-Burke-Ernzerhof exchange-correlation functional with Grimme's DFT-D3 van der Waals correction, as implemented within the Quantum Espresso package[53–55]. A $12 \times 12 \times 6$ Monkhorst-Pack k-point grid was used for the self-consistent calculations and geometry optimisation with a kinetic energy cutoff of 48 Ry for the wavefunctions and 600 Ry for the charge density. We calculated the quasi-particle energies and determine a more accurate estimate of the bandgap using the GW approximation, as implemented within the Yambo package[56,57].

## Data availability

The authors confirm that the data supporting the findings of this study are available within the article and its supplementary information.

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

## Acknowledgements

This work was supported by the European Union's Horizon 2020 research and innovation programme Graphene Flagship Core 3; the Engineering and Physical Sciences Research Council (Grant No. EP/T019018/1) and the University of Nottingham Propulsion Futures Beacon. The high magnetic field studies were supported by the European Magnetic Field Laboratory (EMFL) and by the EPSRC via the UK membership of the EMFL (Grant No. EP/X020304/1).

## Author contributions

A.P., Z.Y. and M.T.G. conceived the project; A.P. wrote the paper; J.O. and J.F. conducted the electron spectroscopy, microscopy and Raman studies with input from A.P.; N.A. conducted Raman and electron spectroscopy studies of samples of different thickness; Z.Y. and Y.K. conducted the magneto-transmission experiments and analysed the data with input from L.E. and A.P.; J.H., J.P., and M.T.G. conducted the DFT studies; all authors discussed the results.

## Competing interests

The authors declare no competing interests.
