## [Transparent Peer Review file · Nature Communications]

Probing and Manipulating the Mexican Hat-Shaped Valence Band of In₂Se₃

Corresponding Author: Professor Amalia Patane

Version 0:

Reviewer comments:

Reviewer #1

(Remarks to the Author)

The primary discovery presented in this manuscript centers on the provision of direct visual experimental evidence grounded by the authors' meticulous nanoARPES measurements to unveil the presence of an inverted Mexican hat-shaped energy band in the valence band of both α and β -phase In₂Se₃. However, in my opinion, these results are not sufficiently striking to meet the novelty criteria for publication in Nature Communications. It is noted that the Mexican hat-shaped bands in In₂Se₃ has been previously predicted by DFT calculations and reported in other 2D materials. I agree with the authors on the point that the Mexican hat-shaped feature typically exists in few-layer samples in other materials while can persist in the bulk of In₂Se₃. But this is not considered to be very critical, particularly for the applications where attempting to push the materials to few-layer or even extreme single-layer limit. If the authors could really demonstrate some correlation manifestations of the Mexican hat-shaped bands, such as charge, magnetic, and superconducting order, as proposed theoretically, the impact of this manuscript would be greatly enhanced. But this links to another issue of this work in the sense that achieving the desired correlation effects require the Fermi level of the system to fall into the Mexican hat-shaped band, but all the samples reported in this manuscripts are n-type. This might pose a big challenge to make such tuning in practice. In addition, there are some other aspects I would suggest the authors to investigate more sophisticatedly. A very distinctive feature of α -In₂Se₃ is its coupled in-plane and out-of-plane electric polarization. This characteristic can lead to various interlayer polarization configurations in its multilayer and bulk forms. On top of that, the different 2H and 3R stacking will further increase the complexity of the dipole alignment. But the present manuscript lacks detailed information on the dipole alignment. It would be valuable for the authors to delve into this aspect and explore how different stacking and interlayer dipole alignment would affect the Mexican hat-shaped bands.

Reviewer #2

(Remarks to the Author)

By using nanoARPES and magneto-optical transmission, in conjunction with DFT calculations, J. Felton et al. demonstrate an inverted Mexican hat-shaped valence band in bulk van der Waals material In₂Se₃. They further show the phase-change from alpha- to beta-In₂Se₃ via a thermal annealing in ultra-high vacuum, serving as a controlled way to tailor the electronic band structure. The study is organized, the results are interesting and the topic would make it suitable for publication in Nature Communications. It may offer a good material platform for studying magnetism, superconductivity, and other phenomena induced by correlation effects. There is however room for improving the manuscript. Specifically, my points are the following:

1. To my knowledge, ferroelectric properties were observed only in ultrathin films of alpha-In₂Se₃ instead of its bulk form. In addition, it was reported that 2H-stacked few-layer alpha-In₂Se₃ exhibits a behavior of layer-dependent ferroelectricity [Mater. Horiz. 8, 1472 (2021)]. Considering that only band dispersion rather than ferroelectric properties in bulk In₂Se₃ is experimentally studied in this manuscript, the used wording, such as "ferroelectric semiconductor" in the title, is inaccurate, which may cause confusion.
2. In the section of Methods, the authors introduce their DFT calculations that have considered spin-orbit coupling. However, the spin splitting caused by the broken inversion symmetry is not observed in most of the calculated band structures. Since Se is a heavy atom, spin-orbit coupling is supposed to have a significant effect on the valence band, thereby affecting the resulting effective masses as well as the position of the inverted Mexican hat.
3. Due to the fact that the ARPES spectrum only cuts a 2D plane out of the 3D BZ, the authors are suggested to scan the out-

of-plane momentum of the photoelectron by varying the photon energy. For comparison, more paths should be sampled in the DFT calculation to show the out-of-plane dispersion in Figure 2.

4. For the calculated band structures of bulk In_2Se_3 , such as Figure 3(a) and Figure S6, the presented results only include the energy dispersion when $k_z=0$. I suggest supplementing the energy dispersion covering all the 3D BZ paths to further demonstrate the conclusions in this work.

Two other minor suggestions:

1. By convention, the abbreviation for density of states is DOS.
2. In Figure 1(c), Figure 4(c,d) and Figure 5, I suggest plotting the 2D BZ.

My recommendation is that the manuscript can be considered for publication in Nature Communications provided that the authors thoroughly address the above points.

Reviewer #3

(Remarks to the Author)

Combining nanoscale angle-resolved photoemission electron spectroscopy and magneto-optical transmission in high magnetic fields to reveal the band structure of the van der Waals ferroelectric semiconductor indium selenide ($\alpha\text{-In}_2\text{Se}_3$). The authors demonstrate the appearance of a Inverted Mexican Hat Valence Band on the surface of a bulk crystal of $\alpha\text{-In}_2\text{Se}_3$. The measurement of this particular crystal phase appears to be novel, and as the authors motivate the combination of ferroelectric behaviour, interesting features in the electronic structure, and the possibility to preserve these features in the few-layer limit is of great technological interest. However, as I explain below, the method for measuring the Mexican Hat valence band using the lower photon energies is not convincing and requires further experimental or theoretical backing, and so I cannot recommend this letter for publication in nature communications. -The Mexican is only visible in the lower photon energy data at 21 eV, but the quality of the ARPES data is not good to conclude about this results. Why is there such a loss of photoemission intensity at this photon energy? The statement that this is due to a reduced cross section at this energy should be supported, if this is the reason. Do the authors have any idea about the quality or oxidation of the sample?

-The measurement of the Mexican Hat valence band uses a different photon energy to determine the respective band positions. This requires the strong assumption that neither of these bands disperse in k_z , which is not well supported. The authors do not explain why the different valence band could not be captured at 21 eV.

-Discussion could made of the k_z dispersion of key bands (in particular the VBM) calculated using DFT. This calculated dispersion should be presented across the full photon energy range measured experimentally, at least in the supplementary material.

Aside from these significant points, there are a couple of more technical questions I would like the authors to address: -The few layers of In_2Se_3 appears to have been measured with photoemission, could the authors provide some comparison to this existing work? -The measurement scales of the micro-Raman used to confirm the crystal phase, and the photoemission are different (oxidation of the sample after the transfer). How has quality sample over the beam size been ensured? -The authors should include some forward looking discussion of possible modifications to the electronic structure with reduced layer thickness, especially as the work was motivated by the interest in the few-layer ferroelectric behaviour of this crystal. - The three refs concerning the presence of a Mexican hat must be cited : <https://www.nature.com/articles/srep05497>, <https://journals.aps.org/prb/abstract/10.1103/PhysRevB.98.115405>, <https://pubs.acs.org/doi/10.1021/acsnano.8b08726>,

Version 1:

Reviewer comments:

Reviewer #1

(Remarks to the Author)

I would like to maintain the main point of criticism from my previous review, namely that the novelty of this work is not sufficient to meet the criteria of Nature Communications. An additional serious concern is that a real demonstration of the correlation effects induced by the Mexican hat-shaped bands may not be practically feasible, as most as-grown In_2Se_3 samples are naturally n-type, and converting them to p-type could be highly challenging. This kind of doping asymmetry is quite common in semiconductors.

The authors stress the importance of SOC, but the DFT calculation results actually show that SOC has a detrimental effect on the Mexican hat-shaped bands. This presentation may confuse readers.

The alpha-phase In_2Se_3 is more stable than the beta-phase, and the phase transition from beta-alpha has been well demonstrated experimentally. Therefore, it may not be appropriate to emphasize this phase transition as 'irreversible'.

In addressing the first comment from Reviewer #2, the layer-dependent ferroelectricity reported in the reference refers to the case where the out-of-plane electric polarization of all layers is ferroelectrically aligned, resulting in layer-dependent in-plane ferroelectricity, or vice versa, due to the coupled out-of-plane and in-plane electric polarizations of alpha- In_2Se_3 and its particular 2H stacking configuration.

Finally, the authors should avoid using unnecessary abbreviations, such as vHs, to improve readability.

Reviewer #2

(Remarks to the Author)

The authors answered all my questions satisfactorily. Therefore, I suggest the publication of the manuscript.

Reviewer #1

The primary discovery presented in this manuscript centers on the provision of direct visual experimental evidence grounded by the authors' meticulous nanoARPES measurements to unveil the presence of an inverted Mexican hat-shaped energy band in the valence band of both α and β -phase In_2Se_3 . However, in my opinion, these results are not sufficiently striking to meet the novelty criteria for publication in *Nature Communications*. It is noted that the Mexican hat-shaped bands in In_2Se_3 has been previously predicted by DFT calculations and reported in other 2D materials.

We thank the reviewer for their positive remarks on our nanoARPES study. However, we respectfully dispute that our results “*are not sufficiently striking to meet the novelty criteria for publication in Nature Communications*”. We note that models in the literature report different band structures for the same material system (see for example ref. 18-19-20). Thus, our observation of a Mexican hat-shaped valence band in α - In_2Se_3 and its modification by a structural phase transition provide important test-beds for these models. Also, to improve our understanding of contradicting results in the literature, we have examined the role of spin-orbit coupling (SOC) in shaping the valence band.

To clarify these points, we have revised the text and included additional data (pg. 2-3, 5-6, 13-14).

I agree with the authors on the point that the Mexican hat-shaped feature typically exists in few-layer samples in other materials while can persist in the bulk of In_2Se_3 . But this is not considered to be very critical, particularly for the applications where attempting to push the materials to few-layer or even extreme single-layer limit. The miniaturization of In_2Se_3 offers a myriad of opportunities. However, understanding changes of materials properties from bulk to thin layers is essential to inform further advances. In particular, as discussed in the revised manuscript, SOC has a significant effect on the form of the valence band, which is differently shaped in bulk and thin layers. This is qualitatively different from previous studies of other metal chalcogenide compounds, such as InSe or GaSe . To clarify this point, we have revised the text and included additional data (pg. 5-6, 13-14 and SI).

If the authors could really demonstrate some correlation manifestations of the Mexican hat-shaped bands, such as charge, magnetic, and superconducting order, as proposed theoretically, the impact of this manuscript would be greatly enhanced. But this links to another issue of this work in the sense that achieving the desired correlation effects require the Fermi level of the system to fall into the Mexican hat-shaped band, but all the samples reported in this manuscripts are n-type. This might pose a big challenge to make such tuning in practice. We believe that our present findings are important in their own right as no previous experiments have successfully revealed the valence band of In_2Se_3 , a prerequisite for studies of carrier correlations. Also, the fact that the layers are n-doped, as due to unintentional doping, does not preclude prospects for the experimental observation of correlation phenomena. As the reviewer states, correlations effects could be observed if the Fermi level falls within the valence band. This would require layer doping by a chemical route (intentional doping), which is beyond the scope of this paper. On the other hand, a recent theoretical paper presents this prospect [Wang, Z. et al. *Adv. Electron. Mater.* 10, 2300642 (2024)]. Other approaches, such as photodoping, would also be possible. To clarify this point, we have revised the text and cited the new article (pg. 14).

In addition, there are some other aspects I would suggest the authors to investigate more sophisticatedly. A very distinctive feature of α - In_2Se_3 is its coupled in-plane and out-of-plane electric polarization. This characteristic can lead to various interlayer polarization configurations in its multilayer and bulk forms. On top of that, the different 2H and 3R stacking will further increase the complexity of the dipole alignment. But the present manuscript lacks detailed information on the dipole alignment. It would be valuable for the authors to delve into this aspect and explore how different stacking and interlayer dipole alignment would affect the Mexican hat-shaped bands. In the absence of an external electric field, the two polarization states would be degenerate and not detectable in our experiments. The investigation of the effects of a polarization on the electronic band structure would require an externally applied electric field. On the other hand, we have now conducted new theoretical studies to examine the effect of SOC, which is significant. Thus, we expect a strong influence of an applied electric field on the band structure with opportunities to use electrical currents to generate spin currents and polarization via SOC. To clarify this point, we have revised the text (pg. 5-6, 13-14).

In summary, we believe that our work brings new important insight into the electronic properties of ferroelectric In_2Se_3 , contributing to stimulate further advances in this topical area of solid-state physics.

Reviewer #2

By using nanoARPES and magneto-optical transmission, in conjunction with DFT calculations, J. Felton et al. demonstrate an inverted Mexican hat-shaped valence band in bulk van der Waals material In_2Se_3 . They further show the phase-change from alpha- to beta- In_2Se_3 via a thermal annealing in ultra-high vacuum, serving as a controlled way to tailor the electronic band structure. The study is organized, the results are interesting and the topic would make it suitable for publication in Nature Communications. It may offer a good material platform for studying magnetism, superconductivity, and other phenomena induced by correlation effects. There is however room for improving the manuscript...

My recommendation is that the manuscript can be considered for publication in Nature Communications provided that the authors thoroughly address the above points.

We thank the reviewer for their positive remarks and recommendations, which we address below.

Specifically, my points are the following:

- 1. To my knowledge, ferroelectric properties were observed only in ultrathin films of alpha- In_2Se_3 instead of its bulk form. In addition, it was reported that 2H-stacked few-layer alpha- In_2Se_3 exhibits a behavior of layer-dependent ferroelectricity [Mater. Horiz. 8, 1472 (2021)]. Considering that only band dispersion rather than ferroelectric properties in bulk In_2Se_3 is experimentally studied in this manuscript, the used wording, such as “ferroelectric semiconductor” in the title, is inaccurate, which may cause confusion.** To reflect better the content of our work, we have revised the title to “*Probing and Manipulating the Mexican Hat-Shaped Valence Band of In_2Se_3* ”. However, we have probed the ferroelectric properties of our bulk thin-films crystals ($\alpha\text{-In}_2\text{Se}_3$) by piezoresponse force microscopy (PFM) (see our previous work in *2D Mater.* 8 045020 (2021)). Also, room temperature ferroelectricity was reported in both thin and thick bulk (>100 nm) layers (see, *Nano Lett.* 17, 5508 (2017)). We thank the reviewer for referring us to the layer-dependent ferroelectricity behaviour in thin-layers (< 10 nm) [Mater. Horiz. 8, 1472 (2021)], which we refer to in the revised manuscript.
- 2. In the section of Methods, the authors introduce their DFT calculations that have considered spin-orbit coupling. However, the spin splitting caused by the broken inversion symmetry is not observed in most of the calculated band structures. Since Se is a heavy atom, spin-orbit coupling is supposed to have a significant effect on the valence band, thereby affecting the resulting effective masses as well as the position of the inverted Mexican hat.** We are very grateful for this insightful suggestion. We have now conducted DFT calculations to account for spin-orbit coupling (SOC), which was not included in the original paper. In the revised manuscript, we present the calculated band structure with and without SOC. The SOC is indeed very important in shaping the valence band.
- 3. Due to the fact that the ARPES spectrum only cuts a 2D plane out of the 3D BZ, the authors are suggested to scan the out-of-plane momentum of the photoelectron by varying the photon energy. For comparison, more paths should be sampled in the DFT calculation to show the out-of-plane dispersion in Figure 2.** We have now performed additional calculations of the band structure for different out-of-plane wave vectors. The additional paths sampled by DFT are illustrated in Figure 1,2, 5 and Figure S4, S7 (see also supplementary video). The video illustrates the colour plot of the electron energy versus in-plane k -vector for different out-of-plane k -vectors (k_z). It can be seen that the band structure undergoes only small changes at different k_z going from the centre to the edge of the Brillouin zone. With regard to the ARPES experiment, we use a He discharge UV source at a fixed energy of 21.2 keV (not a variable energy, as done with synchrotron radiation sources).
- 4. For the calculated band structures of bulk In_2Se_3 , such as Figure 3(a) and Figure S6, the presented results only include the energy dispersion when $k_z=0$. I suggest supplementing the energy dispersion covering all the 3D BZ paths to further demonstrate the conclusions in this work.** We have now included the energy dispersion covering all the 3D BZ paths.

Two other minor suggestions:

- 1. By convention, the abbreviation for density of states is DOS.** We have changed DoS to DOS.
- 2. In Figure 1(c), Figure 4(c,d) and Figure 5, I suggest plotting the 2D BZ.** We have now included the 2D BZ in the new Figures 1d and 5b-c.

In summary, we believe that our revisions have helped to improve our manuscript, addressing criticisms and suggestions for improvement by this reviewer.

Reviewer #3

Combining nanoscale angle-resolved photoemission electron spectroscopy and magneto-optical transmission in high magnetic fields to reveal the band structure of the van der Waals ferroelectric semiconductor indium selenide (α -In₂Se₃). The authors demonstrate the appearance of a Inverted Mexican Hat Valence Band on the surface of a bulk crystal of α -In₂Se₃. The measurement of this particular crystal phase appears to be novel, and as the authors motivate the combination of ferroelectric behaviour, interesting features in the electronic structure, and the possibility to preserve these features in the few-layer limit is of great technological interest. However, as I explain below, the method for measuring the Mexican Hat valence band using the lower photon energies is not convincing and requires further experimental or theoretical backing, and so I cannot recommend this letter for publication in nature communications.

We thank the reviewer for their comments and constructive criticisms, which we address below.

The Mexican is only visible in the lower photon energy data at 21 eV, but the quality of the ARPES data is not good to conclude about this results. Why is there such a loss of photoemission intensity at this photon energy? The statement that this is due to a reduced cross section at this energy should be supported, if this is the reason. We would like to clarify that all measurements in the paper were performed with a fixed photon energy of 21.2 eV by using the primary excitation of a He discharge UV source. We do not vary the photon energy but instead measure constant photoelectron kinetic energy slices through the Brillouin zone. Our main finding of an inverted Mexican-hat shaped VB for α -In₂Se₃ can be inferred from the ARPES and DFT data in the new Figure 1d-e, Figure 2c-d-e, supplementary Figure S3 and supporting video. The video illustrates the colour plot of the electron energy versus in-plane k -vector for different out-of-plane k_z -vectors (k_z). It can be seen that the band structure undergoes only small changes at different k_z going from the centre to the edge of the Brillouin zone. Also, the magneto-transmission experiment at high magnetic fields (Figure 3) and their analysis support the main finding of the ARPES study.

Further clarifications are included in response to additional requests for clarification below.

Do the authors have any idea about the quality or oxidation of the sample?

The reviewer raises an important point. Following a long exposure to oxygen and/or moisture, the surface of In₂Se₃ can degrade by forming an oxide and Se-particles (see for example publication in Yan et al. Angew. Chem. Int. Ed. 2023, 62, e202300302). In our experiment, we use freshly exfoliated thin films of In₂Se₃ and all samples are thermally annealed in UHV to create clean surfaces for the ARPES studies. To clarify this point, we have revised the text and referred to the relevant literature on oxidation.

The measurement of the Mexican Hat valence band uses a different photon energy to determine the respective band positions. This requires the strong assumption that neither of these bands disperse in K_z , which is not well supported. The authors do not explain why the different valence band could not be captured at 21 eV.

With regard to the ARPES experiment, we use a He discharge UV source at a fixed energy of 21.2 keV (not a variable energy, as done with synchrotron radiation sources). To assess the effect of the out-of-plane k_z dispersion on the measured bands, we have performed additional calculations of the energy bands at different k_z . We find that the dependence of the energy on k_z validate our experimental findings. In particular, the new supplementary video illustrates the colour plot of the electron energy versus in-plane k -vector for different k_z . It can be seen that the band structure undergoes only small changes at different k_z , going from the centre to the edge of the Brillouin zone. The inverted Mexican-hat shaped valence band is indeed captured experimentally in Figure 1 and 2, which were revised to illustrate better the experimental and theory data.

Discussion could made of the K_z dispersion of key bands (in particular the VBM) calculated using DFT. This calculated dispersion should be presented across the full photon energy range measured experimentally, at least in the supplementary material. We agree that this information is important and we have now included it in the main manuscript (Figure 2) and supporting information (Figure S7). **Aside from these significant points, there are a couple of more technical questions I would like the authors to address: The few layers of In₂Se₃ appears to have been measured with photoemission, could the authors provide some comparison to this existing work?**

The inverted Mexican hat-shaped valence band of In₂Se₃ has not yet been reported in the literature, but attempts have been made to measure it in single layer α - and β -In₂Se₃, and bulk In₂Se₃. The experimental

investigation of these materials can be complicated by the existence of surface defects and/or contamination. For example, defects can induce the formation of a metallic two-dimensional electron gas on the surface, dominating the ARPES signal. In our experiments, we have optimized the preparation of the crystals to create samples with clean surfaces. This approach enabled us to map out the valence band of the crystals and to modify it in a controlled way via a structural phase transition. To clarify this point, we have revised the text on page 3 and cited the relevant literature.

The measurement scales of the micro-Raman used to confirm the crystal phase, and the photoemission are different (oxidation of the sample after the transfer). How has quality sample over the beam size been ensured?

The micro-Raman experiments are conducted in air. A laser beam of in-plane diameter of about 1 micron was used for the excitation of the samples and the Raman spectra were acquired over several spots of the sample to probe the sample uniformity. The Raman spectra are dominated by bulk properties (rather than surface properties) due to the large optical absorption length. In contrast, ARPES is very sensitive to the surface of the layers. Thus, before the experiment, each sample was thermally annealed in UHV to create a clean surface. In ARPES, photoelectrons were collected from a region of $10 \times 20 \mu\text{m}^2$; also, the ARPES experiments were conducted over several areas of the sample to probe its uniformity.

To clarify these points, we have revised the Methods section.

The authors should include some forward looking discussion of possible modifications to the electronic structure with reduced layer thickness, especially as the work was motivated by the interest in the few-layer ferroelectric behaviour of this crystal.

We have now expanded the discussion of the electronic band structure of this layers, which is strongly affected by spin-orbit coupling (pg. 5-6). Also, we have expanded the conclusion, providing a discussion of future developments in this field (pg. 14). This includes a discussion of advanced techniques for growth of materials in UHV to create high-quality layers.

The three refs concerning the presence of a Mexican hat must be cited:

<https://www.nature.com/articles/srep05497>

<https://journals.aps.org/prb/abstract/10.1103/PhysRevB.98.115405>

<https://pubs.acs.org/doi/10.1021/acsnano.8b08726>

We thank the referee for these suggestions. We have included these additional references on a different material system, i.e. GaSe. This is an example of semiconductor with a parabolic valence band dispersion in the bulk limit, but with an inverted Mexican valence band in the few layer limit. There also important differences between GaSe and In_2Se_3 , with the latter revealing stronger spin-orbit coupling effects. These differences are now discussed in the revised manuscript.

In summary, we believe that our further theoretical work and revisions have helped to strengthen our findings. We are not aware of any similar work in the literature. On the other hand, the interest in the area of ferroelectric semiconductors is steadily increasing. Thus, we believe that our results are novel and important to warrant their publication in Nature Communications.

Reviewer #1 (Remarks to the Author):

I would like to maintain the main point of criticism from my previous review, namely that the novelty of this work is not sufficient to meet the criteria of Nature Communications. An additional serious concern is that a real demonstration of the correlation effects induced by the Mexican hat-shaped bands may not be practically feasible, as most as-grown In₂Se₃ samples are naturally n-type, and converting them to p-type could be highly challenging. This kind of doping asymmetry is quite common in semiconductors.

We thank the reviewer for his/her further comments. In response to the criticism of novelty, we note that our findings provide the first important experimental evidence of the electronic band structure of In₂Se₃, a material system of strong interest for fundamental and applied science. We appreciate the concerns of the reviewer about the practical demonstration of correlation effects in as-grown In₂Se₃. This is indeed challenging, but feasible. The reviewer refers to the natural (spontaneous) n-type doping of In₂Se₃ and a doping asymmetry for this material, which would make difficult to dope it p-type (as required to observe correlation effects). However, his/her view maybe based on a traditional view of doping (based on defect thermodynamics under specific growth conditions). In our previous reply, we refer to a recent theoretical work on p-type In₂Se₃ [Wang, Z. et al. Adv. Electr. Mater. 10, 2300642 (2024)]. We now refer to experimental work on p-type In₂Se₃ [Li et al., *Antimony-Doped p-Type In₂Se₃ for Heterophase Homojunction with High-Performance Reconfigurable Broadband Photovoltaic Effect*, Adv. Electr. Mat. 8, 2200665 (2022)]. Alternative approaches, such as photodoping, could also be feasible. Thus, we believe that there are realistic opportunities not only to realize p-type In₂Se₃, but also to exploit it. To clarify this point, we have revised the text and included a new reference.

The authors stress the importance of SOC, but the DFT calculation results actually show that SOC has a detrimental effect on the Mexican hat-shaped bands. This presentation may confuse readers. The SOC tends to flatten the energy dispersions of the valence band, thus making less prominent the Mexican hat-shape. However, the Mexican hat-shaped valence band is still seen by our DFT and experiment in bulk In₂Se₃. To clarify this point, we have revised the text.

The alpha-phase In₂Se₃ is more stable than the beta-phase, and the phase transition from beta-alpha has been well demonstrated experimentally. Therefore, it may not be appropriate to emphasize this phase transition as 'irreversible'.

Although the phase transition from α to β has been reported, the crystallization behaviour of In₂Se₃ under different conditions (e.g. temperature, pressure, growth conditions) can vary. The β -phase of In₂Se₃ has been proposed as a pseudo-centrosymmetric β_{pc} -phase due to the random off-center positions of the Se-atoms. The random displacement of the Se-atoms can create an effective energy barrier that prevents a reversible transformation of the crystal from β to α . For example, it has been reported that β -In₂Se₃ is stable in thin films or nanosheets at room temperature [Tao and Gu, *Crystalline–Crystalline Phase Transformation in Two-Dimensional In₂Se₃Thin Layers*, Nano Lett. 13, 3501 (2013)] and that the β phase can persist in bulk crystals at room temperature in non-oxidative environments [Ignacio et al., *Air-Stable Atomically Encapsulated Crystalline-Crystalline Phase Transitions in In₂Se₃*, Adv. El. Mat. 10, 2300457 (2024)]. To clarify this point, we have revised the text and added two references.

In addressing the first comment from Reviewer #2, the layer-dependent ferroelectricity reported in the reference refers to the case where the out-of-plane electric polarization of all layers is ferroelectrically aligned, resulting in layer-dependent in-plane ferroelectricity, or vice versa, due to the coupled out-of-plane and in-plane electric polarizations of alpha-In₂Se₃ and its particular 2H stacking configuration.

We thank the reviewer for addressing this point (as discussed in the reference cited in our paper).

Finally, the authors should avoid using unnecessary abbreviations, such as vHs, to improve readability.

We have now revised the text to avoid the use of the vHs abbreviation.

Reviewer #2 (Remarks to the Author):

The authors answered all my questions satisfactorily. Therefore, I suggest the publication of the manuscript.

We thank the reviewer for the positive appraisal of our work and constructive comments that have helped to improve our manuscript.